# Conserved features of TERT promoter duplications reveal an activation mechanism that mimics hotspot mutations in cancer

Carter J. Barger [1,12], Abigail K. Suwala[1,2,3,12], Katarzyna M. Soczek[4],
Albert S. Wang [1], Min Y. Kim[1], Chibo Hong[1], Jennifer A. Doudna[4,5,6,7,8,9],
Susan M. Chang[1,10], Joanna J. Phillips [1,10,11], David A. Solomon [10,11] &
Joseph F. Costello [1,10] ✉

Mutations in the *TERT* promoter represent the genetic underpinnings of tumor cell immortality. Beyond the two most common point mutations, which selectively recruit the ETS factor GABP to activate *TERT*, the significance of other variants is unknown. In seven cancer types, we identify duplications of wildtype sequence within the core promoter region of *TERT* that have strikingly similar features including an ETS motif, the duplication length and insertion site. The duplications recruit a GABP tetramer by virtue of the native ETS motif and its precisely spaced duplicated counterpart, activate the promoter and are clonal in a *TERT* expressing multifocal glioblastoma. We conclude that recurrent *TERT* promoter duplications are functionally and mechanistically equivalent to the hotspot mutations that confer tumor cell immortality. The shared mechanism of these divergent somatic genetic alterations suggests a strong selective pressure for recruitment of the GABP tetramer to activate *TERT*.

Limitless replicative potential is one of the hallmarks of cancer cells[1], enabling malignant transformation. This cellular immortalization can be achieved by reactivating telomerase, an enzyme that maintains telomere length during cell division. Genetic alterations that lead to activation of expression in its subunit *Telomerase Reverse Transcriptase* (*TERT*) are often found in cancer[2–8]. In fact, *TERT* promoter (*TERT*p) mutations are the most common noncoding mutations in human cancers[9] including the majority of glioblastomas[10]. The two *TERT* hotspot mutations G228A and G250A in the core promoter region are annotated by their genomic coordinate, chr5:1,295,228 and

chr5:1,295,250 (human genome build, hg19), respectively. Lower frequency recurrent *TERT*p mutations consist of T161G, G228T, GG228AA and GG242AA[4,11]. *TERT*p mutations can also be annotated by their position relative to *TERT* ATG or translation start, for example −124C > T (G228A). The G228A and G250A *TERT* mutations create new E26 transformation-specific (ETS) binding sites (CCGGAA) and recruit the GA-binding protein (GABP) transcription factor (TF) complex to activate *TERT*[12–15]. The GABP TF complex is an obligate multimer consisting of the DNA binding subunit GABPA and the transactivation subunit GABPB1[16]. Depending on the isoform of the GABPB1 subunit,

[1]Department of Neurological Surgery, University of California, San Francisco, CA, USA. [2]Department of Neuropathology, Institute of Pathology, Heidelberg University Hospital, Heidelberg, Germany. [3]Clinical Cooperation Unit Neuropathology, German Cancer Research Center (DKFZ), German Consortium for Translational Cancer Research (DKTK), Heidelberg, Germany. [4]Department of Molecular and Cell Biology, University of California, Berkeley, CA, USA. [5]Gladstone Institute of Data Science and Biotechnology, Gladstone Institutes, San Francisco, CA, USA. [6]Department of Chemistry, University of California, Berkeley, CA, USA. [7]Molecular Biophysics & Integrated Bioimaging Division, Lawrence Berkeley National Laboratory, Berkeley, CA, USA. [8]Howard Hughes Medical Institute, University of California, Berkeley, CA, USA. [9]Innovative Genomics Institute, University of California, Berkeley, CA, USA. [10]UCSF Helen Diller Family Comprehensive Cancer Center, San Francisco, CA, USA. [11]Department of Pathology, University of California, San Francisco, CA, USA. [12]These authors contributed equally: Carter J. Barger, Abigail K. Suwala. ✉e-mail: joseph.costello@ucsf.edu

the complex either forms a heterotetramer containing GABPA-GABPB1L or heterodimer consisting of GABPA-GABPB1S. In the case of the mutant *TERT*p, the generation of a new ETS motif is positioned so that it is in helical phase with the native ETS motif and the distance between the two motifs is ideal for GABP heterotetramer binding[12].

The *TERT*p is subject to other alterations in cancer such as insertion/deletion mutations, amplifications and epigenetic modifications[17–21]. The significance of many of these alterations, and their potential mechanism of telomerase reactivation is unknown. In this regard, two recent case reports observed an intriguing 22 bp duplication of unknown function within the *TERT*p in two glioblastomas and a thyroid cancer[22,23].

Here, we identified 21 tumor samples of 7 different cancer types with 21–25 bp duplications of wildtype sequence within the core promoter. This sizeable cross-cancer cohort allowed us to discover that the sequence, length and insertion position of the duplication are remarkably conserved, identical in many cases, strongly suggesting a pathogenic role. Furthermore, these observations raised a hypothesis which we test to explain why the features are conserved and mechanistically how they compare to hotspot mutations. Together our results point to the apparent positive selection of wildtype sequence duplications that, like hotspot mutations, recruit the GABP tetramer to activate *TERT*.

## Results

### Duplications in the core *TERT* promoter occur in multiple cancer types and have conserved features

Recurrent point mutations in the *TERT*p that generate de novo ETS binding sites recruit the GABP heterotetramer TF complex (GABPB1L and GABPA) to activate telomerase expression (Fig. 1a, b)[24]. In contrast, in cells with the wildtype *TERT*p GABP binding is not detectable and GABP knockdown does not affect *TERT* expression. In our cancer panel sequencing of glioblastomas, we identified a tumor lacking either of the hotspot point mutations but instead harboring a duplication of wildtype sequence with the core *TERT*p. Although the duplication was deemed a variant of unknown significance, we found intriguing similarity to three duplications reported in two case studies[22,23]. To begin to understand the frequency and features of ETS motif containing *TERT*p duplications in cancer, we surveyed the literature along with AACR Project Genomics Evidence Neoplasia Information Exchange (GENIE) and UCSF500 Pan-Cancer datasets. Most of the AACR GENIE *TERT*p sequencing data were generated by the MSK-Impact targeted sequencing panel. Although the MSK-Impact publication captured variants in the *TERT*p[3], hotspot mutations were the main focus. Subsequently, additional tumors have been profiled by MSK-IMPACT and are included in the GENIE cohort, v10[25]. The UCSF500 Pan-Cancer dataset is available to UCSF affiliates and was generated by the UCSF500 targeted sequencing panel, which covers the *TERT*p. Table 1 includes 21 duplications from 18 cases we identified that are contained within the core region of the *TERT*p, including six samples with duplications from four patients in our cohort. Comparing the studies in Table 1, *TERT*p duplication variants are represented by cancer types with the highest frequencies of the hotspot *TERT*p mutations, which include bladder cancer, glioblastoma, squamous cell carcinoma of the oropharynx, hepatocellular carcinoma and thyroid cancer. *TERT*p duplications are also present in rare forms of breast and intestinal cancer. Comparing patient demographics, *TERT*p duplications are represented by different age groups, sex, and races (Table 1). Together these data show that *TERT*p duplications are not restricted to a specific patient demographic (age or sex) or tumor type (primary site, metastasis or recurrent tumor), but they do tend to occur in the same cancer types as those with frequent *TERT*p hotspot mutations suggesting they may serve a similar function. Although *TERT*p duplications and mutations occur in similar cancer types, broadly, *TERT*p duplications occur at a lower frequency (Supplementary Fig. 1a, Supplementary Table 1).

We next carefully compared the *TERT*p duplication variants in Table 1 to identify any unifying features that might give clues to their function and made two striking observations. Irrespective of cancer type, duplications have an approximate size of 22 bp and contain an exact copy of one of the native ETS motifs, specifically ETS 200 (Table 1). To determine if the ETS containing *TERT*p duplications in Table 1 had unique features as compared to non-ETS duplications in the *TERT*p, we mapped all MSK-IMPACT duplication variants within a 4 kb region upstream of *TERT* (chr5:1295160-1299000) and observed a broad spectrum of insert sizes with no obvious bias for their insert position (Supplementary Fig. 1b). However, when specifically mapping these variants by presence or absence of the GGAA motif, 13 of the 14 GGAA containing insertions were localized within the core promoter adjacent to the native ETS motif (chr5:1295180-1295199) and approximately 22 bp in length. However, the 19 non-GGAA containing insertions were highly variable in length and dispersed throughout the *TERT*p, which suggests that ETS containing duplications have a selective pressure for their length and insertion point (Supplementary Fig. 1b). Interestingly, the one GGAA containing duplication variant that did not map directly adjacent to the native ETS (c.-125_-106) is different than the others as it is further upstream in the *TERT*p and the ETS motif is actually a copy of the G228A hotspot mutation (Table 1; Supplementary Fig. 1b, c). In agreement, this patient was also diagnosed with a G228A mutation. We identified one additional tumor that showed both *TERT*p G228A hotspot mutation and *TERT*p duplication containing the wildtype sequence. Although G228A and G250A hotspot mutations are considered mutually exclusive, rare cases with both mutations have been reported as well.

Regarding the unique features of duplication variants (Table 1), we previously reported the favored features of the GABP heterotetramer TF complex binding for *TERT*p activation as having two ETS binding sites in phase such that their spacing has a periodicity of 10.5 bp (full helical turns), a preference we also observed genome wide[12]. The distance of 22 bp between the duplicated de novo ETS motif and native ETS motif is approximately 2 full helical turns, similar to the distance from the native ETS to either hotspot mutation, putting these two motifs in phase for potentially optimal GABP heterotetramer binding (Fig. 1a–c; Supplementary Fig. 1c).

### *TERT* promoter activation requires the ETS containing duplication and GABP

The wildtype *TERT* promoter is incapable of significant GABP tetramer recruitment, and yet the duplicated sequences are wildtype, leading to uncertainty about how the duplications might activate *TERT* in vivo, and indeed whether they do. However, the precise positioning of the newly acquired ETS motif and its position suggested to us that *TERT*p duplications might function like *TERT*p hotspot mutations to recruit the GABP tetramer to activate transcription. In support, two of the 22 bp *TERT*p duplications previously reported and included in Table 1 increase activity of the *TERT*p in a promoter-reporter assay. Based on sequence motif analysis of the duplication, over one hundred potential TF were predicted that could play a role in *TERT* activation[22,23]. From the conserved features in our relatively large number of cases and in our prior studies, we hypothesized that any 22 bp *TERT*p duplication that specifically includes the native ETS motif could increase promoter activity and the increase would be dependent on insertion point and GABP. To test this hypothesis, we generated twenty luciferase reporter constructs covering all possible 22 bp *TERT*p duplication variants containing the native ETS motif and performed luciferase assays using bladder cancer (UMUC3) and glioblastoma (LN229) cell lines (Fig. 1). We observed that all duplications of the native ETS motif were sufficient for increased *TERT*p activity, at a level very similar to G228A and G250A hotspot mutations (Fig. 1d, e). Interestingly, only duplications containing a copy of native ETS 200, not ETS 195, were sufficient for increased promoter activity. Specifically, duplications

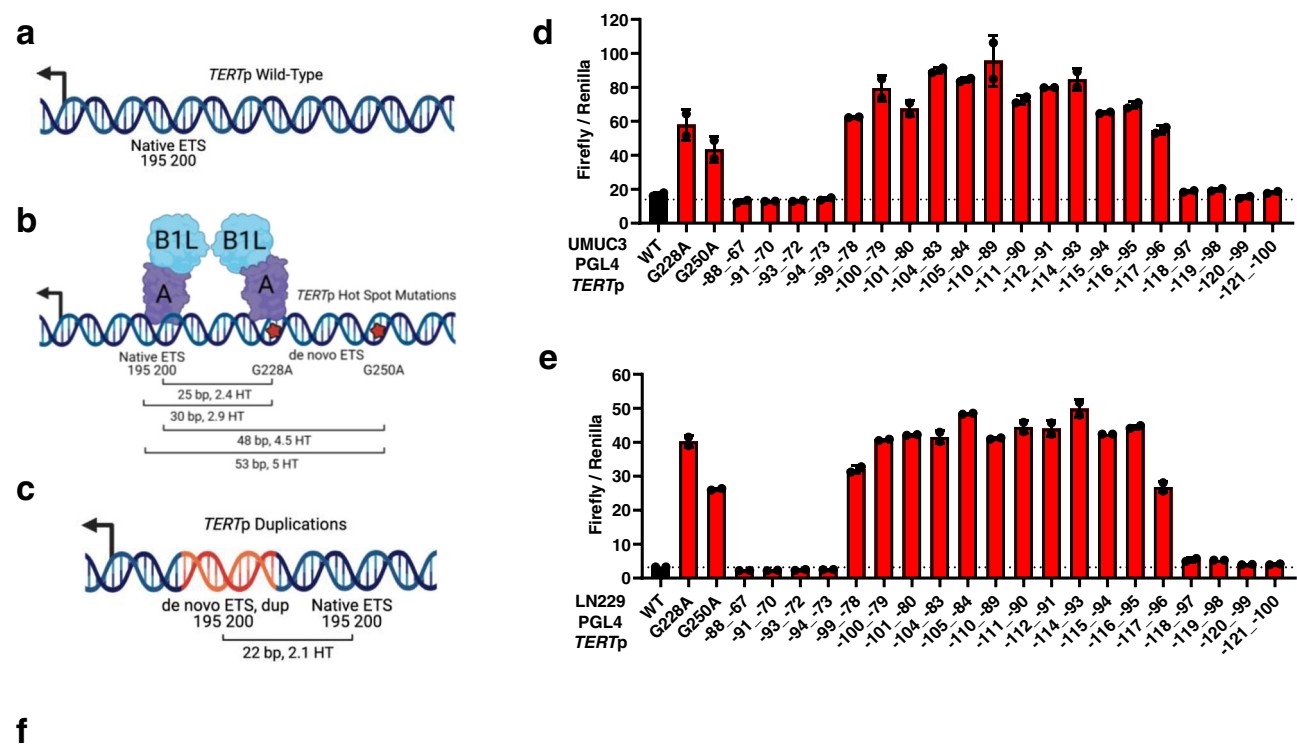

containing only a copy of native ETS 200 (-115_-94 to -117_-96) showed increased reporter activity when compared to the *TERT* wildtype promoter while duplications containing only a copy of native ETS 195 (-94_-73 to -98_-77) did not (Fig. 1d–f). These data agree with Table 1 where ETS 200 is present in all duplicated sequences while ETS 195 is not.

*TERT*p duplication variants are approximately 22 bp in length or 2 full helical turns, which puts the duplicated de novo ETS motif and native ETS motif in phase with each other for GABP tetramer binding (Supplementary Fig. 1c). To determine if this distance of 22 bp between the duplicated de novo and native ETS motif is essential for promoter activity, insertion and deletion mutations were performed to alter the spacing of the c.-100_-79 duplication variant (Supplementary Fig. 2a). We performed luciferase assays using bladder cancer and glioblastoma cell lines and observed that *TERT*p activity was similar between c.-100_-79 and the G228A hotspot mutant (Supplementary Fig. 2b, c). However, promoter activity was abolished as bases were either inserted or deleted from this duplication sequence (Supplementary Fig. 2b, c).

**Fig. 1 | *TERT* promoter activation requires the additional ETS binding site provided by a wildtype sequence duplication or a mutation. a** *TERT*p wildtype schematic. Overlapping native ETS motifs are shown relative to the *TERT* ATG translational start site. **b** *TERT*p hotspot mutation schematic. Native ETS motif and de novo ETS motifs (hotspot mutations) are shown with GABP tetramer binding to the G228A mutation and native ETS motif along with the relative position of the *TERT* ATG. Distance (bp) and number of helical turns (HT) between the native ETS motif and de novo motifs (hotspot mutations) are shown. **c** *TERT*p duplication schematic. Native ETS motif and de novo ETS motif (duplication) are shown relative to the *TERT* ATG. Distance (bp) between the native ETS motif and de novo ETS motif (duplication) is shown along with the associated helical turns (HT). The figures were created with BioRender (**a**–**c**). **d** *TERT*p luciferase reporter assays for wild type, hotspot mutants and duplications at 24 h after transfection in UMUC3 bladder cancer cells. Values are mean and SD. $N = 2$ independent experiments. **e** *TERT*p

luciferase reporter assays for wildtype, hotspot mutants and duplications at 24 h in LN229 glioblastoma cells. Values are mean and SD. $N = 2$ independent experiments. **f** All possible 22 bp *TERT*p duplications spanning the native ETS motif and beyond are shown along with their DNA sequence including the duplication insertion point in bold, location of de novo and native ETS motifs are in red and green font, respectively, and associated reporter activity from Fig. 1d, e. The following duplication sequences generate identical variant sequence upon insertion into the *TERT*p: **−88_−67** = −89_−68, −90_−69; **−91_−70** = −92_−71; **−94_−73** = −95_−74, −96_−75, −97_−76, −98_−77; **−101_−80** = −102_−81, −103_−82; **−105_−84** = −106_−85, −107_−86, −108_−87, −109_−88; and **−112_−91** = −113_−92. Thus, all 20 unique duplications were interrogated with reporter assays while all 34 duplication sequences are shown in panel f for reference and comparison. Abbreviations: *TERT*p, *TERT* promoter. Source data are provided as a Source Data file.

Interestingly, *TERT*p activity increased again as bases were inserted and the distance between de novo and native ETS motifs was around 30 base pairs or approximately 3 helical turns, which put the ETS binding sites in phase of each other again (Supplementary Fig. 2b, c). Although these motifs were in phase, the activity never reached that of c.-100_-79, which suggests the distance of -2 helical turns may be ideal for *TERT*p duplications. The results across this large collection of engineered variants of the duplicated promoter were consistent between the two cancer cell lines with substantially different mutational profiles, transcription factor expression patterns and epigenetic states.

The shared features between the *TERT*p duplication and hotspot mutations suggested to us that GABP may play a functional role in the activation of the *TERT*p duplication. To determine if the *TERT*p duplication is dependent on GABP for the increase in promoter activity, we treated UMUC3 and LN229 cells with control and GABPA siRNAs to deplete the GABP TF complex then performed luciferase assays with these *TERT*p duplication sequences. Similar to our observations in Fig. 1d, e, all ETS containing duplications showed increased reporter activity when compared to the *TERT* wildtype promoter. In stark contrast, upon knockdown of GABPA, we observed a dramatic reduction to wildtype *TERT*p levels for all ETS containing duplications in both cell lines, and this level of reduction was similar to G228A and G250A hotspot mutations (Fig. 2a, b). Next, we investigated whether GABP can bind to the fragment of the *TERT*p with the 22 bp insertion in an electromobility shift assay. Our results show GABP binds to sequences with duplicated native ETS sites (Fig. 3). Together, these data show that GABP is an essential regulator of the *TERT*p duplication, mimicking the hotspot mutations in activation strength and mechanism[12,13].

## Clonality of the *TERT* promoter duplication in a multifocal glioblastoma

*TERT*p hotspot mutations are sufficient for the diagnosis of glioblastoma in IDH-wildtype diffuse astrocytic gliomas according to the latest 2021 WHO Classification of Central Nervous System Tumors[26]. In contrast, the diagnostic value of *TERT*p duplications is unknown. We identified a patient with a *TERT*p duplication whose clinical history and tumor histological and molecular profile were consistent with an integrated diagnosis of glioblastoma, IDH-wildtype and for which intratumoral samples were available for clonality analysis (Fig. 4a). The patient presented with two months of disorientation, double vision, right-sided vision loss and neglect, as well as difficulty with fine motor movements. Imaging analysis showed a multifocal enhancing mass in the left parieto-occipital lobes (Fig. 4b, c). Histology revealed a diffuse astrocytic glioma with pleomorphic tumor cells, brisk mitotic activity, microvascular proliferation, and palisading necrosis (Fig. 4d, e). The patient was treated with radiation and temozolomide, and experienced tumor recurrence after 13 months (Supplementary Fig. 3a, d).

From a total of 13 independent punches from five FFPE tissue blocks of the two portions of the newly diagnosed tumor, we co-purified genomic DNA and total RNA. We first performed PCR on the DNA to determine if the *TERT*p duplication is present in a subset of these samples suggesting sub-clonality, or if it is detectable in all available samples, indicative of an evolutionarily early and clonal event. The duplicated allele resolved as a higher molecular band on the agarose gel as compared to the wildtype allele (Fig. 4f). The detection in all thirteen DNA samples further highlights the early evolutionary origin and clonality of the *TERT*p duplication in this case. We also detected a small amount of the duplication PCR product in the "normal" tissue sample, which was brain parenchyma immediately adjacent to the tumor that likely contained a small population of infiltrating tumor cells.

To determine when the *TERT*p duplication was acquired during tumor evolution relative to other alterations, we performed exome sequencing on three different locations of the occipital and two different locations of the parietal primary tumor, as well as patient-matched blood cell DNA. We determined the SNV, copy number and tumor purity estimates and then used the SNV to generate a sample-oriented phylogenetic reconstruction and a complementary tumor clone-based analysis using the PyClone algorithm[27]. Gain of chromosome 7, loss of chromosome 10 and homozygous deletion of the *CDKN2A* locus were confirmed in all analyzed tumor parts, as was the *TERT*p duplication and an *NF1* frameshift mutation (Fig. 4g, Supplementary Figs. 3b, c, 5a–e). The *TERT*p duplication is present throughout the parietal and occipital portions of the tumor that were analyzed, similar to the distribution of the characteristically clonal alterations (chromosome 10 loss and chromosome 7 gain). In contrast to the clonal events that form the trunk of the phylogenetic tree, sub-clonal driver events were also readily apparent and underlie the tree branches. For example, only the occipital tumor region showed amplification of the *EGFR* locus, whereas a *PTPN11* missense mutation and *ERRFI1* homozygous deletion were exclusively in the parietal tumor region (Fig. 4g, Supplementary Figs. 3b, 5a–e). PyClone analysis allowed us to infer the clonal population structure and revealed different gene mutation clusters in different parts of the multifocal tumor, except for cluster 1 which is present in all parts representing fully clonal, or "truncal" mutations that we infer to have arisen very early in the evolution of this multifocal tumor. Cluster 1 from the PyClone analysis and the trunk of the phylogenetic tree contain a shared set of 17 mutations which, along with the clonal copy number changes, may represent the ancestral clone that seeded the occipital and parietal portions of this tumor. Cluster 2 includes the *PTPN11* mutation and was present only in the parietal tumor focus while cluster 3 which includes *EGFR* amplification was only detected in the occipital tumor focus (Fig. 4h). Like cluster 3, clusters 4 and 5 were also confined to the occipital lobe but appeared to be less abundant. Panel sequencing of a low tumor purity sample of the recurrent tumor revealed a newly acquired *PTEN* tandem duplication in exon 4

**Table 1 | Tumor samples with an ETS binding site duplication in the *TERT* promoter**

| Duplicated Region[1] | Duplicated Sequence | Insert Size (bp) | Insert Position[5] Start | End | Other Acquired ETS Variants | Age | Race | Sex | Cancer Type | Tumor Type | Dataset |
|---|---|---|---|---|---|---|---|---|---|---|---|
| c.-100_-79 | GGGCGGGGCCCGCGGAAAGGAAG | 22 | 1295182 | 1295183 | NA | NA | NA | NA | Glioblastoma Multiforme | NA | Pierini et al. |
| c.-100_-79[2] | GGGCGGGGCCCGCGGAAAGGAAG | 22 | 1295182 | 1295183 | None | 54 | NA | Male | Glioblastoma Multiforme | NA | UCSF500 |
| c.-100_-79 | GGGCGGGGCCCGCGGAAAGGAAG | 22 | 1295182 | 1295183 | G228A | 29 | NA | Female | Squamous Cell Carcinoma | Metastasis | UCSF500 |
| c.-100_-79 | GGGCGGGGCCCGCGGAAAGGAAG | 22 | 1295182 | 1295183 | None | 68 | White | Male | Duodenal Adenocarcinoma | Metastasis | GENIE |
| c.-100_-79[3] | GGGCGGGGCCCGCGGAAAGGAAG | 22 | 1295182 | 1295183 | None | 48 | NA | Male | Glioblastoma Multiforme | Primary | UCSF500 |
| c.-100_-79[3] | GGGCGGGGCCCGCGGAAAGGAAG | 22 | 1295182 | 1295183 | None | 48 | NA | Male | Glioblastoma Multiforme | Primary | UCSF500 |
| c.-100_-79[3] | GGGCGGGGCCCGCGGAAAGGAAG | 22 | 1295182 | 1295183 | None | 48 | NA | Male | Glioblastoma Multiforme | Recurrent | UCSF500 |
| c.-104_-83 | GGGGCCGCGGAAAGGAAGGGGGA | 22 | 1295186 | 1295187 | None | 42 | Asian | Male | Oral Cavity Squamous Cell Carcinoma | Metastasis | GENIE |
| c.-104_-83 | GGGGCCGCGGAAAGGAAGGGGGGA | 22 | 1295186 | 1295187 | None | 50 | White | Male | Bladder Urothelial Carcinoma | Primary | GENIE |
| c.-104_-83 | GGGGCCGCGGAAAGGAAGGGGGGA | 22 | 1295186 | 1295187 | None | 68 | Black | Male | Hepatocellular Carcinoma | Primary | GENIE |
| c.-104_-83 | GGGGCCGCGGAAAGGAAGGGGGGA | 22 | 1295186 | 1295187 | NA | NA | NA | NA | Papillary Thyroid Cancer | NA | Panebianco et al. |
| c.-110_-89 | GCGGAAAGGAAGGGGGAGGGGGCT | 22 | 1295192 | 1295193 | None | 48 | Asian | Female | Malignant Phyllodes Tumor of the Breast | Primary | GENIE |
| c.-110_-89 | GCGGAAAGGAAGGGGGAGGGGGCT | 22 | 1295192 | 1295193 | None | 64 | White | Male | Glioblastoma Multiforme | Primary | GENIE |
| c.-110_-89 | GCGGAAAGGAAGGGGGAGGGGGCT | 22 | 1295192 | 1295193 | None | 65 | NA | Male | Glioblastoma Multiforme | Primary | GENIE |
| c.-110_-89[4] | GCGGAAAGGAAGGGGGAGGGGGCTA | 23 | 1295192 | 1295193 | None | 69 | White | Female | Papillary Thyroid Cancer | Metastasis | GENIE |
| c.-110_-89[4] | GCGGAAAGGAAGGGGGAGGGGGCTA | 23 | 1295192 | 1295193 | None | 72 | White | Female | Poorly Differentiated Thyroid Cancer | Metastasis | GENIE |
| c.-110_-89 | GCGGAAAGGAAGGGGGAGGGGGCT | 22 | 1295192 | 1295193 | NA | NA | NA | NA | Glioblastoma Multiforme | NA | Pierini et al. |
| c.-116_-96 | GGAAGGGGAGGGGCTGGGAGG | 21 | 1295199 | 1295200 | None | 64 | White | Male | Hepatocellular Carcinoma | Primary | GENIE |
| c.-120_-96 | GGAAGGGGAGGGGCTGGGAGGCCC | 25 | 1295199 | 1295200 | None | 80 | NA | Female | Poorly Differentiated Thyroid Cancer | NA | UCSF500 |
| c.-120_-96 | GGAAGGGGAGGGGCTGGGAGGCCC | 25 | 1295199 | 1295200 | None | 68 | White | Male | Hepatocellular Carcinoma | Primary | GENIE |
| c.-125_-106 | GGGGCTGCGGCTGGGAGGCCCGGAA | 25 | 1295228 | 1295229 | G228A | 69 | White | Male | Bladder Urothelial Carcinoma | Metastasis | GENIE |

[1]Duplicated region of the *TERTp* relative to TERT translation start site (ATG);
[2]SF5681.
[3]Primary and recurrent tumors for the same patient, SF12747 and SF13241;
[4]Two metastatic tumors for the same patient;
[5]Human reference genome hg19, chromosome 5; De novo native ETS 195 motif is in bold while de novo native ETS 200 motif is underlined, and de novo ETS motif is in italic. Bold and italic nucleotides in the duplication sequence indicate a mismatch when compared to the duplication template sequence. NA Not available.

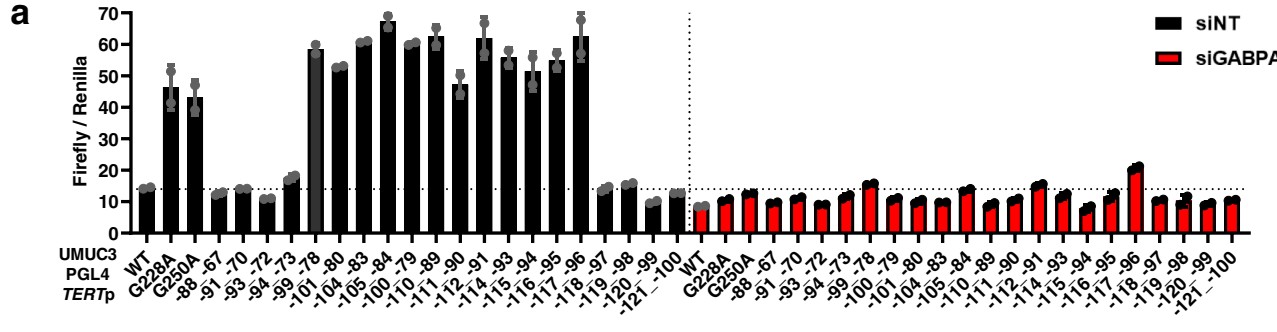

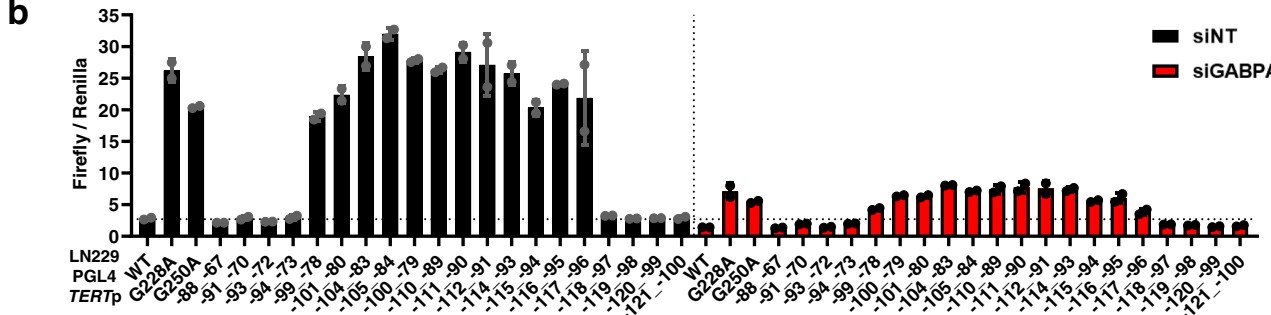

**Fig. 2 | *TERT* promoter duplication confers GABP dependency by contributing a de novo ETS factor binding site. a** *TERT*p luciferase reporter assays for wildtype (WT), two hotspot mutants and 20 unique duplications 72 h after siRNA knockdown of control or GABPA in UMUC3 cells. **b** *TERT*p luciferase reporter assays for wildtype, two hotspot mutants and 20 duplication constructs representing 17 unique sequences 72 h after siRNA knockdown of control or GABPA in LN229 cells. Values are mean and SD. $N = 2$ independent experiments. *TERT*p *TERT* promoter, NT Non-targeting. Source data are provided as a Source Data file.

and retention of the 22 bp *TERT*p duplication (Supplementary Fig. 3d). These somatic genetic data show a common clonal origin of the spatially divergent tumor, with the *TERT*p duplication being among the earliest alterations and being retained during tumor recurrence.

We identified a second glioblastoma case from the UCSF500 cohort with *TERT*p duplication, gain of chromosome 7, loss of chromosome 10 and homozygous deletion of *CDKN2A* as well as an *NF1* frameshift mutation (Supplementary Fig. 4). We did not have sufficient material to perform further analyses on this tumor sample.

### Similar strength of *TERT* activation by duplication and hotspot mutations

*TERT*p hotspot mutations recruit the GABP tetramer complex to activate transcription. Whether the *TERT*p duplications are passenger alterations or can drive *TERT* reactivation in tumors is unknown. Based on the increased promoter activity we observed for ETS containing *TERT*p duplications (Figs. 1, 2), we hypothesized that *TERT*p duplications activate *TERT* expression in vivo. We were unable to identify a human cell line with a *TERT*p duplication, therefore, to determine the association between *TERT*p duplication and *TERT* expression, we used the *TERT*p duplication (c.-100_-79) glioblastoma sample presented in Fig. 4. *TERT* expression is limited to stem cells and cancer cells, therefore, to selectively measure *TERT* expression in tumor cells from this *TERT*p duplicated glioblastoma, we performed *TERT* RNA in situ hybridization (RNAscope) on FFPE slides. Quantification of the RNAscope staining for these *TERT*p duplicated glioblastoma cells broadly shows low level nuclear expression of *TERT* mRNA in individual cells that is comparable to hotspot G228A mutation glioblastoma tissue (Fig. 5a, b, d)[28]. To prove specificity of the assay, we assayed an IDH-mutant astrocytoma with wildtype *TERT*p and detected no *TERT* mRNA expression (Fig. 5c, d). *TERT* mRNA positive cells were observed in a

second glioblastoma case with a *TERT*p duplication (Supplementary Fig. 6).

Next, we measured *TERT* mRNA by RT-qPCR and like the RNAscope result, *TERT* was expressed in the tumor tissue but not in the adjacent "normal" FFPE punches and greater than or equal to FFPE punches obtained from a separate glioblastoma case with the hotspot mutation G228A (Fig. 5e). We also measured the mRNA expression of the GABP TF complex subunits, *GABPB1* and *GABPA*, which shows detectable and robust expression in all available tumor regions (Fig. 5f, g). These data show that the *TERT*p duplication associates with transcriptional reactivation of *TERT*, and in this glioblastoma which also has uniform expression of the mutant *TERT*p transcriptional regulator, GABP. Together, these data suggest that the duplications are likely drivers of *TERT*p reactivation and *TERT* expression in vitro and in vivo.

## Discussion

*TERT*p hotspot mutations are the most common non-coding mutations in human cancers and selectively recruit the GABP TF complex[9]. In contrast, one case report of duplication in the *TERT*p predicted greater than one hundred candidate transcription factors solely based on sequence motifs in the duplication. Considering only ETS factors as candidates, the matter is also complicated by the largely shared binding site preference among the 29 family members. Through a pan-cancer data analysis and our tumor samples, we discovered key shared features of the duplications that allowed insight into their mechanism of promoter activation and likely pathogenic function, leading us to the GABP tetramer. The duplications have similar and even identical length and insertion site across at least 7 cancer types, including duodenal adenocarcinoma in which *TERT*p mutations are rare. The duplications mimic pathogenic hotspot mutations in their transcriptional activation of the *TERT*p, the rate limiting step in reactivating

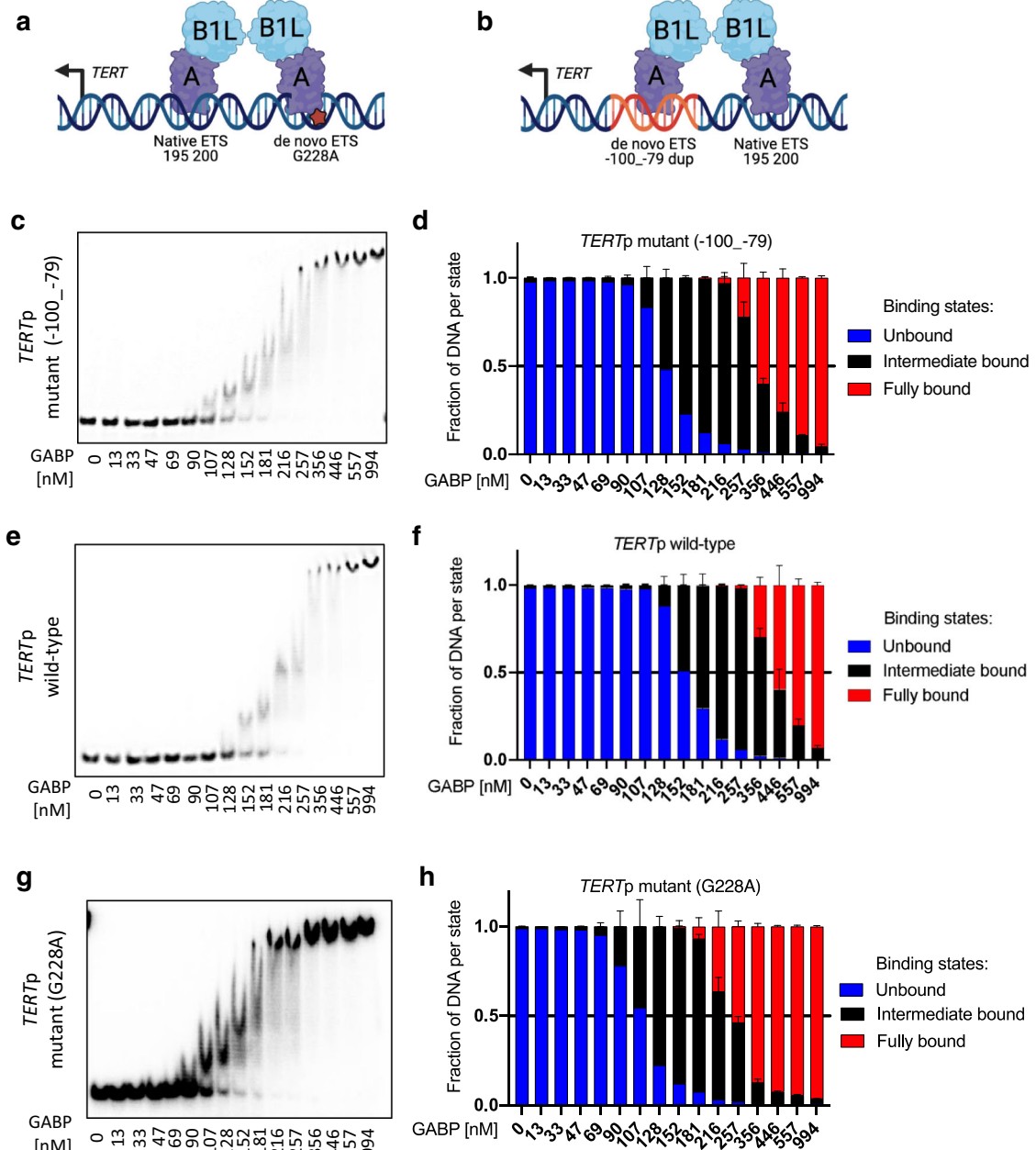

**Fig. 3 | GABPB1L containing GABP complexes bind the *TERT* promoter duplication.** **a**, **b** Schematic of the GABP heterotetramer TF complex binding to the de novo and native ETS motifs of the *TERT*p with the G228A mutation or c.−100_−79 duplication, respectively. The figures were created with BioRender. Representative gels (**c**, **e**, **g**) and quantification (**d**, **f**, **h**) of EMSAs comparing binding affinity of GABPA-B1L heterodimers to the *TERT*p c.−100_−79 duplication, *TERT*p wildtype (native ETS motif only) and G228A mutant (native and de novo ETS motifs) DNA fragments. Abbreviations: *TERT*p, *TERT* promoter. Source data are provided as a Source Data file. Values are mean and SD. *N* = 3 technical replicates.

telomerase activity and cellular immortalization, and in the mechanism by which they do so. Although the hotspot mutations primarily recruit GABP, further studies will be needed to assess exclusivity of GABP recruitment by the duplications. On the other hand, knockdown of GABP alone is sufficient to rapidly reduce *TERT* expression from the promoter with either hotspot mutation or from each of the many different promoters with an ETS-containing duplication.

The location and size of GGAA containing duplications in the *TERT*p suggested to us that they create an additional binding site to enable GABP tetramer binding, representing another genetic mechanism by which the GABP TF complex is recruited to the *TERT*p. In fact, we showed that the GGAA motif and GABP TF complex are both essential for increased *TERT*p duplication activity. The length of the

*TERT*p duplication puts the two ETS motifs in phase and at a distance that is most ideal for tetramer binding (Fig. 6). We observed that as the distance between the ETS motifs increases, the activation of the *TERT*p duplication is decreased. The requirement for two ETS motifs in helical phase further supports GABP as a prime regulator of the promoter duplication because GABP, specifically the tetramer, is the only member of the ETS family that requires two binding motifs.

*TERT*p hotspot mutations are considered one of the early events in glioblastoma formation[29,30]. Likewise, we found *TERT*p duplication in all five spatially independent regions and in a total of 13 punches from the five FFPE blocks of the newly diagnosed tumor and also at recurrence. We infer the *TERT*p duplication is fully clonal in this case and is retained during tumor progression. Therefore, like the hotspot

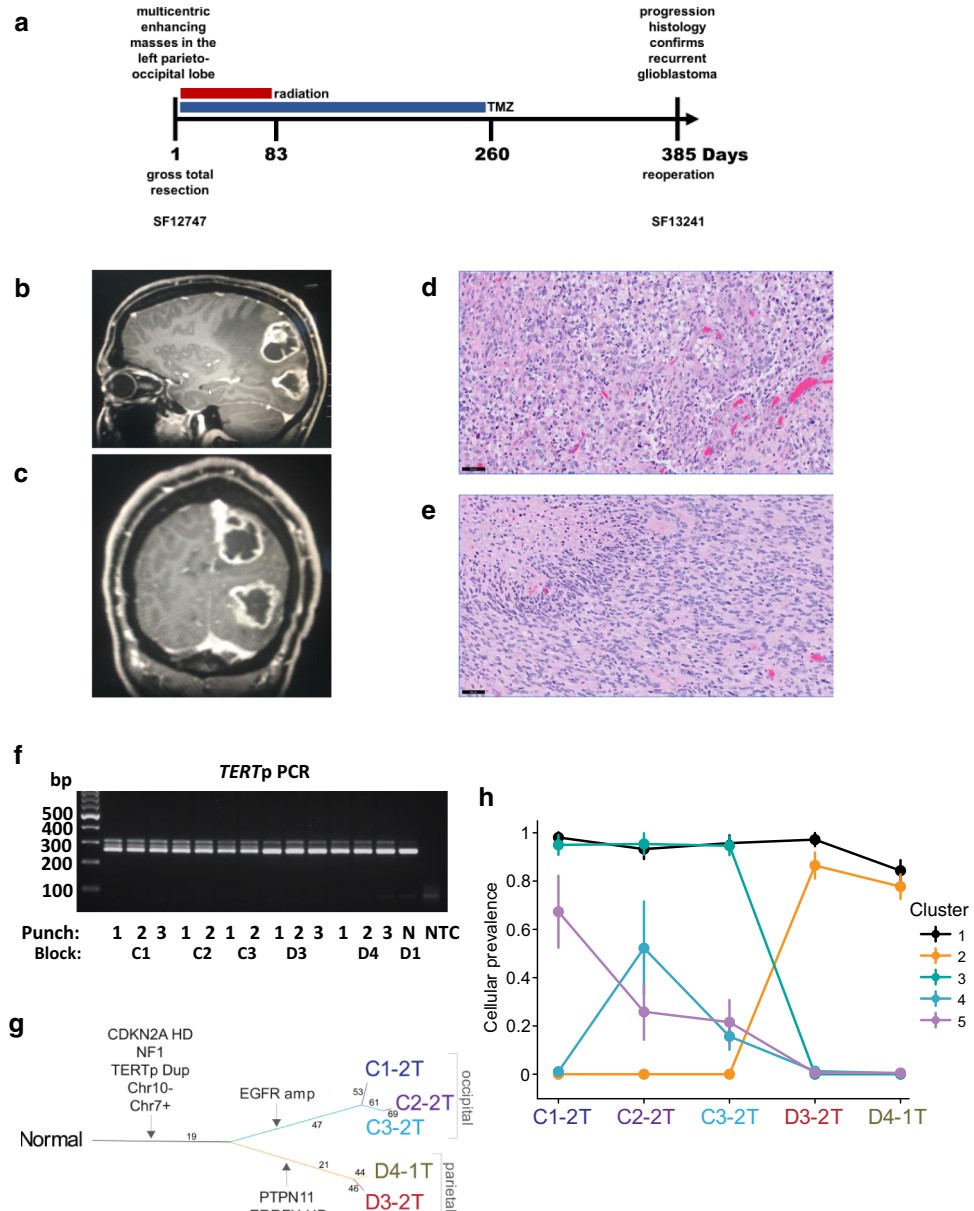

**Fig. 4 | Clonal evolution of a multifocal glioblastoma with the *TERT*p duplication. a** Clinical timeline. Multiple tumor tissue samples were obtained from the IDH-wildtype and *TERT*p duplicated glioblastoma at diagnosis (SF12747) and one sample was obtained at recurrence (SF13241). **b**, **c** MRI of enhancing masses in the left parieto-occipital lobes with surrounding edema and mass effect. **d**, **e** H&E staining of an FFPE section from the (**d**) occipital lobe tumor focus and the (**e**) parietal lobe tumor focus. Scale bar denotes 50 μm. **f** *TERT*p PCR spanning the native ETS motif and duplication for the DNA isolated from FFPE punches from multiple regions of the newly diagnosed tumor, and a no-template control (NTC). Lane 1 includes the DNA ladder for reference of base pair (bp) size. C# and D# indicate the block, # indicate the tumor and N the tumor-adjacent normal brain punch, respectively. The experiment was performed once because of the limitation of patient material. **g** SNV-based phylogenetic tree from the exome sequencing of

DNA from five regions of the tumor and patient-matched normal peripheral blood. Terminal nodes (bold text) represent samples, where N indicates the Normal germline (peripheral blood); all other terminal nodes are tumor samples. Edges (lines) are proportional to the genetic distance between samples within each patient, with the number of mutations corresponding to each major edge indicated. **h** PyClone analysis of the mutated gene clusters from the exome sequencing of DNA from five regions of the tumor relative to patient-matched normal peripheral blood. $N = 1$ biologically independent samples for each region. Error bars indicate the mean s.d. (using 10,000 post-'burn-in' samples) of Markov chain Monte Carlo−derived cellular prevalence estimates over all mutations in a cluster as defined by PyClone algorithm. *TERT*p *TERT* promoter, HD Homozygous deletion, amp Amplification, Dup Duplication. Source data are provided as a Source Data file.

---

mutations, *TERT*p duplication may also be an early event in tumorigenesis, adding to their remarkable similarities despite being fundamentally different somatic genetic alterations.

Interestingly, we identified two cases (one squamous cell and one bladder urothelial carcinoma) with *TERT*p duplication and a *TERT*p hotspot mutation. As there is limited access to these sequencing data, it remains unclear whether the duplication and hotspot mutation were on different alleles within the same cell or in different cells within the

tumor. We identified one tumor with both *TERT*p hotspot mutation and duplication of the mutant sequence, suggesting that the duplication may have provided additional selective advantage in this highly proliferative tumor type, though further experimentation is needed. For the second tumor, the *TERT*p duplication harbored the wildtype sequence. As reads were low for both the *TERT*p duplication and *TERT*p hotspot mutation in this case these could potentially be two concurrent subclonal events that formed independently from each other

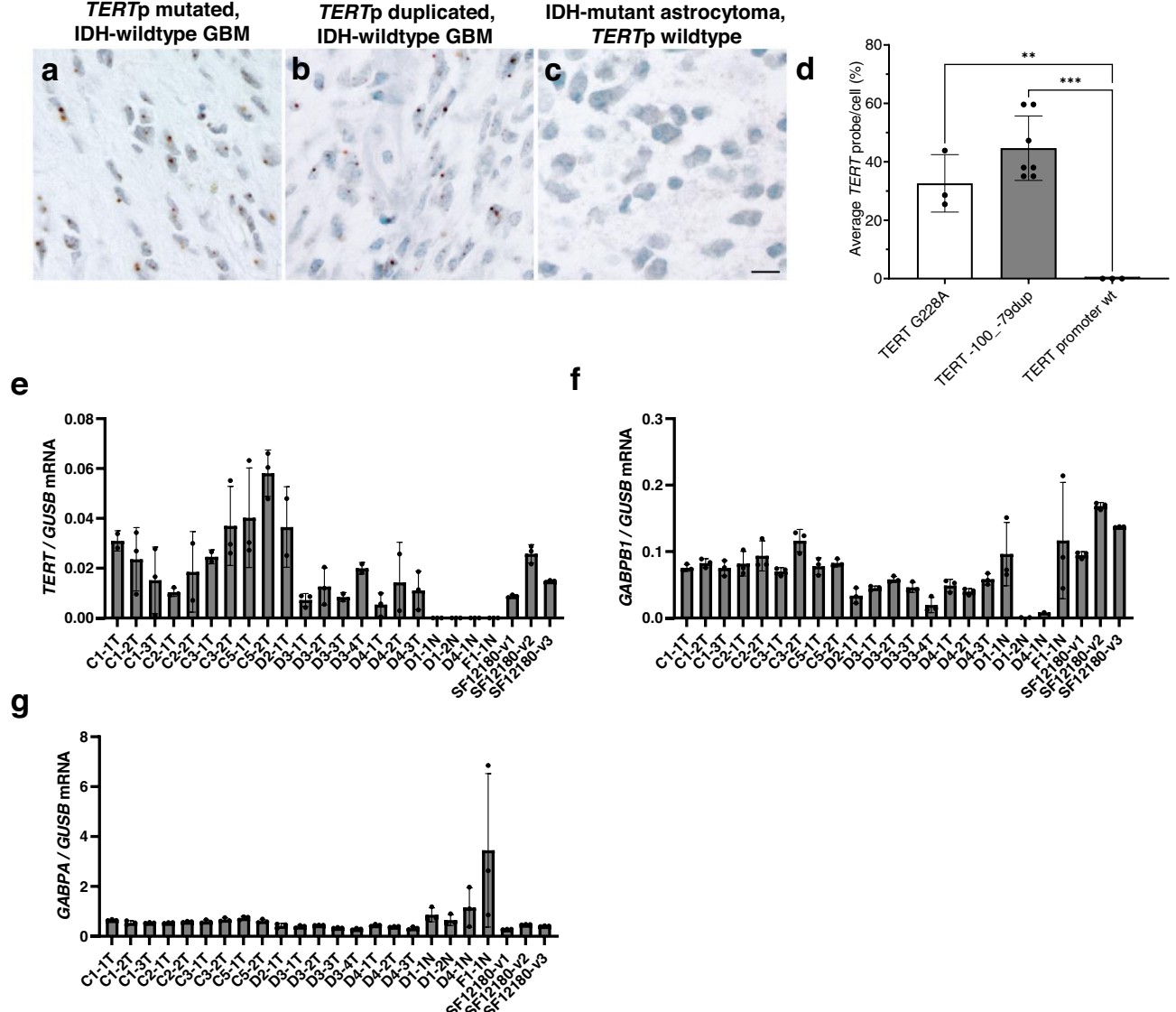

**Fig. 5 | Similar *TERT* expression in glioblastomas with *TERT* promoter duplication or hot-spot mutation. a–d** RNAscope for *TERT* expression in human glioblastoma. Representative images from RNAscope demonstrating **(a)** *TERT* mRNA detected at the single cell level in a positive control glioblastoma, IDH-wildtype with *TERT* G228A mutation, **b** *TERT* mRNA in glioblastoma, *IDH*-wild type with *TERT* −100_−79 duplicated (SF12747), **(c)** *TERT* mRNA in a negative control IDH-mutant astrocytoma with *ATRX* mutation and wildtype *TERT*p. Nuclei stained with hematoxylin. Scale bar denotes 30 µm **(a–c)**. **d** Quantification of the average *TERT* probe per total cell number (%) for the 3 tumors illustrated in **(a–c)**. $N = 1496$ cells examined over 3 independent experiments for TERT G228A, $n = 3204$ cells

examined over 7 independent experiments for TERT −100_−79dup, $n = 547$ cells examined over 3 independent experiments for TERT promoter wt. **(e–g)** *TERT*, *GABPB1* and *GABPA* RT-qPCR results normalized to *GUSB* and performed on RNA isolated from the multifocal *TERT*p duplicated tumor (T) (SF12747), adjacent normal (N) (SF12747) and *TERT*p G228A (SF12180) FFPE punches. Values are mean and SD. Multiple comparisons were performed using 1-way ANOVA, and post hoc analyses were based on Tukey's test. \*\*$p = 0.005$, \*\*\*$p = 0.0001$. C/D# indicates the FFPE tissue block and T/N# indicates the punch. Abbreviations: *TERT*p, *TERT* promoter. Source data are provided as a Source Data file. $N = 3$ technical replicates.

in different cells. Similarly, rare cases of co-occurring G228A and G250A have been reported. Nevertheless, most tumors have only one hotspot mutation or the core promoter duplication.

Our results are also important for clinical diagnosis and tumor classification. For diagnostic purposes, discovering new *TERT*p mutations and distinguishing pathogenic mutations from variants of unknown significance is important since the 2021 WHO Classification of Central Nervous System Tumors specifies histologically low grade diffuse astrocytic gliomas without *IDH*-mutation but with *TERT*p mutations shall be diagnosed as CNS WHO grade 4 glioblastomas[26]. There are many more examples in other tumor types where *TERT* promoter mutations can change grading or diagnosis. These examples include *TERT* promoter status in meningiomas[26], ovarian clear cell

carcinoma[31], small cell carcinoma of the bladder[32], high-grade adenocarcinoma of the prostate[33], conjunctival melanoma[34] and thyroid carcinoma[35]. To date, only *TERT*p hotspot mutations are considered pathogenic. Our data indicates that *TERT*p duplications should also be considered pathogenic for clinical evaluation and tumor classification purposes. This could be implemented immediately, as clinical panel sequencing in use at many academic medical centers and commercial laboratories can routinely detect the duplications. One potential limitation would be for medical centers that use only Sanger sequencing for *TERT*p mutation detection, although even these simple assays could be modified slightly to detect the duplication, as we show in Fig. 4f. For panel sequencing, several of the algorithms used do not detect insertions or deletions greater than 5 bp, and thus the 22 bp

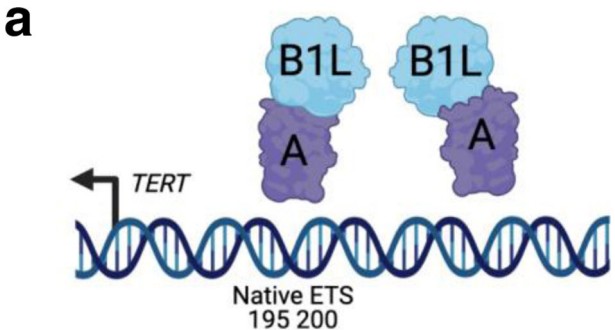

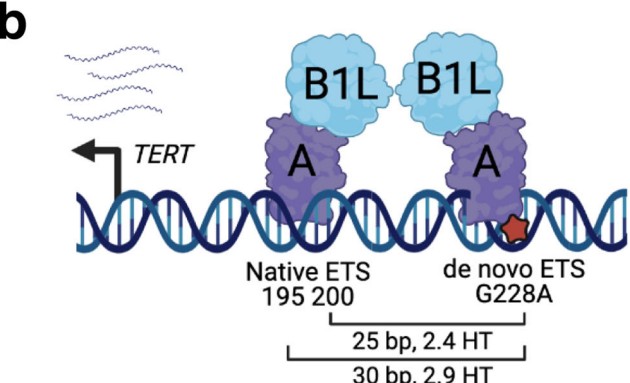

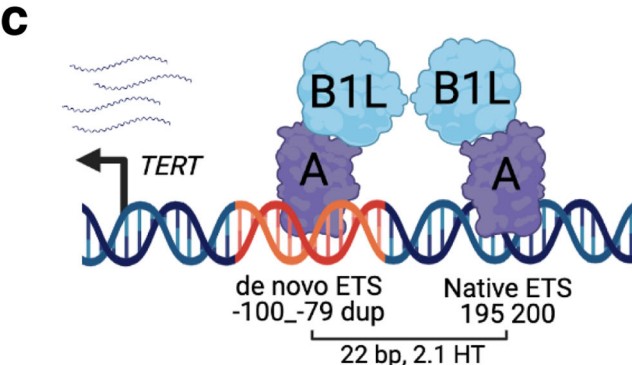

**Fig. 6 | *TERT* promoter duplications mimic hotspot mutations for GABP tetramer recruitment. a** Schematic of the wildtype *TERT*p. The native ETS motif is shown relative to the *TERT* ATG translational start site. GABPB1L-GABPA heterodimer is not bound to the native ETS motif and an inactive *TERT*p. **b** Schematic of the *TERT*p with a hotspot mutation. The native ETS motif and de novo ETS motifs (hotspot mutation) are shown relative to the *TERT* ATG. Distance in base pair (bp) between the native ETS motif and de novo motifs (hotspot mutation) is shown along with the associated helical turns (HT). GABPB1L-GABPA tetramer binding to the native ETS motif and de novo ETS motif (G228A, hotspot mutation) to activate the *TERT*p. **c** Schematic of the *TERT*p duplication. Native ETS motif and de novo ETS motif (c.−100_−79, duplication) are shown relative to the *TERT* ATG. Distance (bp) between the native ETS motif and de novo ETS motif (duplication) is shown along with the associated helical turns (HT). GABPB1L-GABPA tetramer binding to the native ETS motif and de novo ETS motif (c.−100_−79, duplication) to activate the *TERT*p. The figures were created with BioRender (**a**–**c**).

*TERT*p duplications may be missed. This potential issue could be further exacerbated by the consistent observation that promoter regions with high GC-content such as the *TERT*p yield lower sequencing depth[36,37]. Our findings in Supplementary Table 1 might not be representative, as *TERT*p mutations were only detected in one third of all glioblastomas in the GENIE dataset whereas multiple independent

studies (including our own experience as part of the UCSF Glioblastoma Precision Medicine Program) suggest the frequency is over 80%. It is therefore possible that *TERT*p duplications are underreported in cancer.

Our results show that *TERT*p duplications are recurrent, pathogenic variants across multiple cancer types, and begin to define their frequency, size, insertion site bias and functional significance. GABP is the common denominator between *TERT*p duplications and hotspot mutations, supporting a key role in cellular immortality. The remarkable parallel between genetically distinct hotspot mutations and core promoter duplications together suggest a selective pressure specifically for GABP tetramer recruitment to reawaken one allele of the epigenetically silenced *TERT*p. Understanding the characteristics of the GABP tetramer and its binding partners may shed light on this apparently unique capability.

## Methods

This study complies with all relevant ethical regulations and was approved by the Committee on Human Research of the University of California, San Francisco. All patients provided informed written consent prior to sample acquisition. Patients were not monetarily compensated.

### AACR project GENIE dataset
The tenth dataset of AACR Project Genomics Evidence Neoplasia Information Exchange (GENIE), Cohort v10.0-public, was accessed from cBioPortal and queried for *TERT* promoter mutations[25]. This dataset was filtered for *TERT*p mutations and insertions containing GGAA. De-identified clinical data such as cancer diagnosis, sex, age, race and the anatomic site of the tumor sequenced were also obtained for the associated cases.

### UCSF500 pan-cancer dataset
The UCSF500 NGS Panel[38] is a panel sequencing assay of 479 genes that are involved in human cancer. The UCSF500 NGS Panel pancancer dataset was accessed from a private cBioPortal instance that is available to UCSF clinicians, researchers, or other staff. The UCSF500 cBioPortal instance currently has de-identified UCSF500 NGS Panel data as well as select de-identified clinical data elements such as diagnosis, sex, and the anatomic site of the tumor sequenced with consent to publish this information. Race was self-defined. This dataset was filtered for *TERT*p mutations and insertions containing GGAA.

### UCSF500 glioblastoma patient cohort and tumor samples
Two patients with glioblastomas harboring duplications within the upstream regulatory region of the *TERT* gene as identified by capture-based next-generation DNA sequencing (UCSF500) performed on a prospective clinical basis as part of the UCSF Glioblastoma Precision Medicine Program were included in this study. All tumor specimens were fixed in 10% neutral-buffered formalin and embedded in paraffin. Tumor tissue was selectively scraped from unstained slides or punched from formalin-fixed, paraffin-embedded blocks using 2.0 mm disposable biopsy punches (Integra Miltex Instruments, #33-31-P/25) to enrich for maximal tumor content. Genomic DNA was extracted from this macro-dissected formalin-fixed, paraffin-embedded tumor tissue using the QIAamp DNA FFPE Tissue Kit (Qiagen, #56404).

### Targeted next-generation DNA sequencing
Capture-based next-generation DNA sequencing was performed using a custom cancer gene panel, UCSF500[38], and an Illumina HiSeq 2500 instrument. This assay targets all coding exons of 479 cancer-related genes, selected introns and upstream regulatory regions of 47 genes. DNA regions captured were also chosen to enable detection of structural variants including gene fusions, and DNA segments at regular intervals along each chromosome to enable genome-wide copy

number and zygosity analysis. The total sequencing footprint is 2.8 Mb. 250 ng of DNA was used for multiplex library preparation using the KAPA Hyper Prep Kit (Roche, #07962347001) according to the manufacturer's specifications. Hybrid capture of pooled libraries was performed using a custom oligonucleotide library (Nimblegen SeqCap EZ Choice). Captured libraries were sequenced as paired-end 100 bp reads. GRCh37 (hg19) was used as the reference genome, and sequenced reads were aligned using the Burrows-Wheeler aligner (BWA). Recalibration and deduplication were performed using the Genome Analysis Toolkit (GATK). Coverage and sequencing statistics were performed using Picard CalculateHsMetrics and Picard CollectInsertSizeMetrics. FreeBayes, Unified Genotyper, and Pindel were used to call single nucleotide variant and small insertion/deletion mutations. Delly was used for large insertion/deletion and structural alteration calling. Variant annotation was performed with Annovar. Variants were visualized and verified using Integrative Genome Viewer. Genome-wide copy number and zygosity analysis was performed by CNVkit and visualized using Nexus Copy Number (Biodiscovery).

### Exome sequencing
Exome capture was performed using xGen Exome Research Panel v2. Tumor cell purity and chromosomal copy number was estimated using FACETS from the exome sequencing data[39]. Exome libraries were sequenced on an Illumina NovaSeq 6000 instrument. The reference human genome build used for all sequencing data alignment in this study was GRCh37 (hg19).

### Phylogenetic tree construction and PyClone analysis
SNVs were called from the exome sequencing of tumor and matching normal genomic DNA by MuTect to generate sample-oriented sSNV-based phylogenetic trees. To generate a phylogenetic tree, an Ordinary lease squares minimum evolution from ape R package was implemented using a distance matrix for all samples from the patient. The Manhattan distance was computed for the binary call matrix, and fasteme.ols from R package ape was used to construct a rooted binary tree[40–42]. PyClone (version PyClone-0.13.1) was also used to perform the clonal frequency analysis by grouping SNVs into clonal clusters[5,27].

### RNAscope
FFPE sections were evaluated by RNAscope chromogenic in situ hybridization (CISH) assay for the expression of *TERT* in individual cells using Advanced Cell Diagnostics (ACD) probes (Newark, CA) specific for *TERT* (481969-C2). The RNA Probe PPIB (313909) and dapB (312039) were used as positive and negative control probes, respectively. Digital photomicrographs were taken using an Olympus UC90 camera with 3-6 images containing >1000 cells evaluated per tumor. Image analysis was performed using QuPath to enumerate total cells per image, based on nuclear hematoxylin staining, and total number of probes per image. The percentage of *TERT* probe/cell from each image was then averaged to determine the average percent *TERT* probe/cell for each tumor.

### Mammalian cell culture
All cell lines were maintained at 37 °C in a humidified incubator with 5% CO$_2$. Cell culture medium was changed every 3–5 days depending on cell density. For routine passage, cells were split at a ratio of 1:3-10 when they reached 80% to 90% confluence. LN229 human glioblastoma cells (ATCC, #CRL-2611) were cultured in Dulbecco's Modified Eagle Medium/Nutrient Mixture F-12 (DMEM/F-12; Corning, #10-090-CV) supplemented with 10% fetal bovine serum (FBS; Gibco) and 100 U/mL Penicillin-Streptomycin (Gibco). UMUC3 human bladder cancer cells (ATCC, #CRL-1749) were cultured in Minimum Essential Medium (MEM; Corning, #10-009-CV) supplemented with 10% FBS (Gibco) and 100 U/mL Penicillin-Streptomycin (Gibco). Cell lines were

authenticated by short tandem repeat (STR) analysis at the University of California Berkeley Sequencing Facility and confirmed to be Mycoplasma free by PCR followed by TBE agarose gel electrophoresis using a previously published method[43]. For mycoplasma testing, cell lines were grown for a period of 72 h in the absence of Penicillin-Streptomycin then 1 mL of culture supernatant was collected and centrifuged at 13,000 x g for 5 min at room temperature. Supernatant was aspirated then the resulting pellet was lysed with 50 μL of Quick-Extract (Lucigen) and incubated on a thermal cycler at 65 °C for 15 min, 68 °C for 15 min and 98 °C for 10 min. A total of 2 μL QuickExtact lysate was used as template in 20 μL PCR reaction containing Phusion Polymerase (Thermo Fisher Scientific) and standard desalted primers. PCR primer sequences: 5_Myco_F1CGCCTGAGTAGTACGTWCGC; 5_Myco_F2 TGCCTGRGTAGTACATTCGC; 5_Myco_F3CRCCTGAGTAGTATGCTCGC; 5_Myco_F4CGCCTGGGTAGTACATTCGC; 3_Myco_R1GCGGTGTGTACAARACCCGA; and 3_Myco_R2GCGGTGTGTACAAACCCCGA. Cycling conditions were: 1 cycle at 98 °C for 30 s, 34 cycles at 98 °C for 10 s, 65 °C for 10 s and 72 °C for 20 s, and a final extension at 72 °C for 5 min. PCR reactions were resolved on a 1.3% TBE agarose gel containing 1X SYBR Safe DNA Gel Stain (Thermo Fisher Scientific). Internal control and positive control DNA were obtained from the DSMZ, German Collection of Microorganisms and Cell Cultures, Braunschweig, Germany (https://www.dsmz.de/fileadmin/user_upload/Collection_MuTZ/Order_form_Internal_Control_neu.pdf). Internal control DNA was spiked into duplicate PCR reactions to rule out a false negative due to PCR inhibition and the positive control was included in each PCR batch. Cell lines were expanded then the stocks were tested for absence of mycoplasma contamination and lack of mycoplasma contamination was confirmed every 6 months.

### *TERT* promoter PCR and genotyping
Genomic DNA was purified from UMUC3 and LN229 cell lines using the Quick DNA Plus Kit (Zymo Research, #D4069). 100 ng of genomic DNA was used for PCR with Phusion Polymerase (Thermo Fisher Scientific, #F530S) containing GC buffer and 5% DMSO and standard desalted primers. PCR primer sequences were designed with tails incorporating M13 primer sequences: M13_*TERT*p_PCR_F 5′-GTAAAACGACGGCCAGACGTGGCGGAGGGACTG-3′ and M13_*TERT*p_PCR_R 5′-CAGGAAACAGCTATGACAGGGCTTCCCACGTGCG-3′. Cycling conditions were: 98 °C for 1 min, 69 °C for 15 s and 72 °C for 30 s for a total of 32 cycles with a final extension at 72 °C for 5 min. A fraction of the PCR product was resolved on a 2% TBE agarose gel containing 1X SYBR Safe DNA Gel Stain (Thermo Fisher Scientific, #S33102) to confirm the size and quality then the remainder was column purified using DNA Clean and Concentrator-5 (Zymo Research, #D4003) and submitted for Sanger sequencing (GENEWIZ) with the standard desalted sequencing primers: M13_seq_F 5′-GTAAAACGACGGCCAG-3′ and M13_seq_R: 5′-CAGGAAACAGCTATGAC-3′.

Glioblastoma tumor tissue (SF12747) was selectively punched from formalin-fixed, paraffin-embedded (FFPE) blocks using 2.0 mm disposable biopsy punches (Integra Miltex Instruments, #33-31-P/25) to enrich for as high of tumor content as possible. RNase-treated genomic DNA was purified from glioblastoma FFPE punches using AllPrep DNA/RNA FFPE Kit (Qiagen, #80234). 100 ng of genomic DNA was used for PCR with Phusion Polymerase (Thermo Fisher Scientific, #F530S) containing GC buffer, 5% DMSO and standard desalted primers (IDT): M13_*TERT*p_PCR_F 5′-GTAAAACGACGGCCAGACGTGGCGGAGGGACTG-3′ and M13_*TERT*p_PCR_R 5′-CAGGAAACAGCTATGACAGGGCTTCCCACGTGCG-3′. Cycling conditions were: 98 °C for 1 m, 69 °C for 15 s and 72 °C for 30 s for a total of 32 cycles with a final extension at 72 °C for 5 min. PCR products were resolved on a 2% TBE agarose gel containing 1X SYBR Safe DNA Gel Stain (Thermo Fisher Scientific, #S33102) and imaged on the UVP UVsolo (Analytik Jena) with a UV wavelength setting of 302 nm.

## Molecular cloning of luciferase reporter plasmids

*TERT* promoter reporter variants were generated by PCR amplification from single stranded DNA Ultramer oligos (IDT) using Phusion Polymerase (Thermo Fisher Scientific, #F530S) and the following standard desalted primers (IDT): XhoI_dup_F 5′-CTAGCCTCGAGTCCTGCCCCTTCACCTTCCAG-3′ and HindIII_dup_R 5′-TTGCCAAGCTTGGCCGCCGAGGCCAGATCCAGCGCTGCCTGAAA-3′. PCR products were gel purified using Zymoclean Gel DNA Recovery Kit (Zymo Research, #D4001), digested with XhoI and EcoRI (Thermo Fisher Scientific, #FD0695, #FD0275) then column purified using DNA Clean & Concentrator-5 (Zymo Research, #D4004). Digested fragmented were then ligated into XhoI and HindIII (Thermo Fisher Scientific, #FD0695, #FD0505) linearized pGL4.0-*TERT* WT (Addgene, #84924) plasmid. All PGL4 *TERT*p duplication variants were verified by Sanger sequencing (GENEWIZ).

## Luciferase promoter reporter assay

Transfection of *TERT*p reporter plasmids was carried out with ViaFect transfection reagent (Promega, #E4981). Briefly, cells were seeded at a density of 3000 or 6000 cells per well in a 96-well clear bottom white polystyrene microplate (Corning) plate in antibiotic free media. 24 h post-seeding, cells were transfected with 90 ng pGL4.10 plasmid (experimental promoter with Firefly luciferase, Promega), 9 ng pGL4.74 plasmid (control promoter with Renilla luciferase; Promega), and 0.3 µL of ViaFect transfection reagent (Promega, #E4981) in 10 µL of Opti-MEM serum free media (Gibco). Non-targeting (#D-001206-13-20) and GABPA (#M-011662-01) directed siGENOME SMART siRNA pools (Dharmacon) were used for knockdown experiments. For knockdown experiments, cells were seeded at a density of 2,000 or 5,000 cells per well in a 96-well plate in antibiotic free media. 24 h post-seeding cells were transfected with 30 nM of siRNA and 0.3 µL of DharmaFECT reagent (Dharmacon, #T-2001-02) in 10 µL of Opti-MEM serum free media (Gibco). At 48 h post-transfection, cells were transfected with reporter plasmids as described above. 24 h post transfection of reporter plasmids, Firefly luciferase activity was measured by using the Dual-Luciferase Reporter assay system (Promega, #E1960) on the Promega GloMax 96 well microplate luminometer and normalized against Renilla luciferase activity then presented as a ratio of Firefly to Renilla using arbitrary units.

## Protein purification

Genes for GABP β1L and GABP α were cloned to pET-Duet-1 vector using Gibson assembly. GABP β1L was inserted after the first ribosome binding site and GABP α after the second ribosome binding site. GABP β1L has a N-terminal His-Tag separated from the subunit by TEV cleavage sequence.

Subunits were expressed in Rosetta DE3 cell line. A single colony was used to inoculate an overnight starter culture in Luria-Bertani media (LB) with 100 µg/ml Ampicillin. Next, starter culture was used to inoculate large Terrific Broth (TB) media cultures with 100 µg/ml Ampicillin. These cultures were grown at 37 °C until the $OD_{600}$ reached 0.8-1 at which point they were chilled on ice and induced with 0.5 mM IPTG. Protein expression was conducted at 16 °C overnight. Cells were harvested via centrifugation, resuspended in 50 mM HEPES pH 7 500 mM NaCl, 1 mM TCEP, 5% (v/v) glycerol and flash frozen. For protein purification pellets were thawed and mixed with 0.5 mM PMSF and cOmplete protease inhibitor (Roche) and lysed by sonication. Following centrifugation, cleared lysate was incubated for 30 min on the rocker at 4 °C with Ni-NTA Superflow resin (Qiagen). Resin was pre-equilibrated with 50 mM HEPES pH 7, 500 mM NaCl, 1 mM TCEP, 5% glycerol, 10 mM imidazole buffer. After incubation resin was washed with 200 ml of the equilibration buffer. Protein elution was performed with 50 mM HEPES pH 7, 500 mM NaCl, 1 mM TCEP, 5% (v/v) glycerol, 500 mM imidazole buffer. Fractions containing both subunits were combined and dialysed overnight in Slide-A-Lyzer (ThermoFisher) with

addition of TEV protease in 50 mM HEPES pH 7.5, 150 mM NaCl, 1 mM TCEP, 5% (v/v) glycerol. Next subunits were purified through Ni HiTrap HP (GE) and Heparin HiTrap column (GE) using dialysis buffer as buffer A and 50 mM HEPES pH 7.5 1 M KCl, 1 mM TCEP, 5% (v/v) glycerol as eluting buffer B. Fractions with both subunits were combined, concentrated and loaded on Superdex 200 Increase 10/300 equilibrated with 20 mM HEPES pH 7.5, 200 mM KCl, 1 mM TCEP, 5% (v/v) glycerol. Purified protein was concentrated, flash frozen and stored at −80 °C.

## Electrophoretic mobility shift assay

Single stranded 60 bp oligonucleotides with wild type *TERT*p including native ETS sites (5′-AGGGCGGGGCCG**CGGAAAGGAA**GGGGAGGGGCTGGGAGGGCCCGGAGGGGGCTG GGCCGG-3′), mutant *TERT*p with native and de novo ETS sites created by G228A mutation (5′-AGGGCGGGGCCG**CGGAAAGGAA**GGGGAGGGGCTGGGAGGGCC**CGGAaG** GGGCTGGGCCGG-3′) and duplication mutant creating a new de novo site (5′-AGGGCGGGGCCG**CGGAAAGGAA**GGGGCGGGGCC G**CGGAAAGGAA**GGGGAGGGGCTGGGAGGGCCCGGAGGGGGCTGGGCCGG-3′) were obtained from IDT as PAGE purified. Then one of the strands of each sequence was labeled with $^{32}$P and annealed to its complementary unlabeled strand in 20 mM Tris pH 7.5, 100 mM LiCl, 1 mM $MgCl_2$ buffer and used for the EMSA assay.

To introduce $^{32}$P label, 10 pmols of single stranded DNA were incubated with 5 Units of T4 PNK (NEB) and 10µCi of $^{32}$P-γ-ATP (Perkin-Elmer) in PNK buffer in total volume of 25 µl for 30 min at 37 °C and heat inactivated at 65 °C for 20 min. To remove excess of ATP, reaction was diluted by addition of 25 µl of DEPC-treated water and applied to a G25 (GE) desalting column. For annealing, purified oligos were further diluted by addition of 50 µl of DEPC-treated water.

To perform EMSA assays, GABP proteins were thawed and diluted to starting stock concentration of 2 µM and then further serial dilutions were performed in Superdex 200 buffer to create a series of sixteen total protein stocks. Annealed DNA oligonucleotides were diluted to 1 nM stock concentration in annealing buffer. Reactions were performed by mixing DEPC-treated water, 5x reaction buffer and DNA to create a master stock, which was then equally distributed between reaction tubes and supplemented with protein from sixteen stocks. The final reaction component concentrations were as follows: 0.2 nM 60 bp DNA, GABPA/B heterodimers (0 to ~1000 nM) with final concentration of reaction buffer 20 mM Tris pH 7.5, 50 mM KCl, 5.2 mM $MgCl_2$, 1 mM TCEP, 5% (v/v) glycerol in 15 µl final reaction volume. Reactions were incubated at room temperature for 1 h and then 3 µl were applied to a 5% (v/v) native 0.5X TBE PAGE gel with 5 mM $MgCl_2$, which was prerun at 8 W for ~1 h at 4 °C. The gel was then run in 0.5X TBE buffer with 5 mM $MgCl_2$ for ~2.5 h at 8 W at 4 °C. The gel was dried and phosphorous screen was exposed to the gel overnight. Imaging was performed using a Typhoon phosphorimager and quantification with ImageQuant software. The fraction of promoter sequence bound was calculated by dividing the signal of bound DNA fraction by the sum of bound and unbound signals in each lane.

## Reverse transcription quantitative PCR (RT-qPCR)

Glioblastoma tumor tissue (SF12747) was selectively punched from formalin-fixed, paraffin-embedded (FFPE) blocks using 2.0 mm disposable biopsy punches (Integra Miltex Instruments, #33-31-P/25) to enrich for maximal tumor content. Tissue punches of adjacent normal brain parenchyma microscopically devoid of tumor cells were also generated for comparison. DNase-treated total RNA was purified from glioblastoma and adjacent normal FFPE punches using AllPrep DNA/RNA FFPE Kit (Qiagen, #80234). Briefly, 0.5 µg of DNase-treated RNA was converted to cDNA using the iScript cDNA Synthesis Kit (Bio-Rad, #1708891). cDNA was diluted 1:2 using nuclease free water, and 2 µL of diluted cDNA was used for qPCR reactions with POWER SYBR Green PCR Master Mix (Thermo Fisher Scientific, #4367659). Standard curves were prepared using gel-purified end-point RT-PCR products.

All samples were run in triplicate using the QuantStudio 5 (Thermo Fisher Scientific) and all gene expression data were normalized to *GUSB* mRNA. PCR was performed with an annealing temperature of 60 °C and a total of 45 cycles was run for all primer pairs. Dissociation curves were performed to confirm specific product amplification. RT-qPCR standards for each gene were generated from a mixture of human cell line cDNA via end-point RT-PCR then gel purification, using the appropriate primer pair. Gradient PCR reactions were performed with the C1000 Touch Thermal cycler (Bio-Rad) to determine the annealing temperature and specificity for each primer set. Primer sequences corresponding to each gene for the mRNA expression analysis were designed using NCBI Primer Blast or selected from those previously reported in the literature: *TERT*_RT_F 5′-TCACGGAGAC CACGTTTCAAA-3′; *TERT*_RT_R 5′-TTCAAGTGCTGTCTGATTCCAAT-3′; GABPB1_RT_F 5′-AAACGGGTGTATCTGCTGTTC-3′; GABPB1_RT_R 5′-GGCCACTACTGGAGTTTCTGAA-3′; GABPA_RT_F 5′-AAGAACGCCTTG GGATACCCT-3′; GABPA_RT_R 5′-GTGAGGTCTATATCGGTCATGCT-3′ GUSB_RT_F 5′-CTCATTTGGAATTTTGCCGATT-3′; and GUSB_RT_R 5′-CCGAGTGAAGATCCCCTTTTTA-3′.

## Statistics & reproducibility

No statistical method was used to predetermine sample size. Sample sizes were chosen based on availability of suitable material. All experiment were done in at least technical duplicates and compared using two different cell lines if suitable. All attempts at replication were successful. No data were excluded from the analyses. The experiments were not randomized. The Investigators were not blinded to allocation during experiments and outcome assessment. Representative images of H&E stained and RNA scope sections from diagnostic tumor specimens are shown. Each tumor specimen was carefully examined by an expert neuropathologist and showed similar findings throughout the resected tumor tissue from which representative images were selected.

## Reporting summary

Further information on research design is available in the Nature Research Reporting Summary linked to this article.

## Data availability

The raw exome sequencing data generated in this study have been deposited in the European Genome-Phenome Archive database under accession code EGAD00001008768. The raw panel sequencing data generated in this study have been deposited in the European Genome-Phenome Archive database under accession code EGAD00001009286. The exome and panel sequencing data are available under restricted access for IRB requirements, access can be obtained by contacting J.F.C. (joseph.costello@ucsf.edu). The GENIE dataset can be accessed via https://genie.cbioportal.org/login.jsp after registration on the platform. The data from Panebianco et al. (ref. 22) are available from the corresponding author upon reasonable request (we used the processed and published data). Pierini et al. (ref. 23) used *TERT* promoter specific PCR and Sanger sequencing and thus the data (sequencing traces) and results are available in their publication. Source data are provided with this paper (including uncropped gels and blots). Source data are provided with this paper.

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

## Acknowledgements

We thank the UCSF BTRC and the staff of the UCSF Clinical Cancer Genomics Laboratory for assistance with genetic profiling. This work was supported by NIH fellowship, NCI K00CA212470 (C.J.B.), NIH grant NCI 2P50CA097257 (J.F.C.), NIH grant NCI P50CA097257 (J.F.C. and J.A.D.), funding from the UCSF Glioblastoma Precision Medicine Program (J.F.C. and J.A.D.), a generous gift from the Dabbiere family (J.F.C.), and a generous gift from the Hana Jabsheh Research Initiative (J.F.C.). This study was supported in part by the Panattoni Family Foundation and the UCSF Glioblastoma Precision Medicine Program sponsored by the Sandler Foundation. D.A.S. is supported by the NIH Director's Early Independence Award from the Office of the Director, National Institutes of Health (DP5 OD021403). A.K.S is supported by the Mildred Scheel program from the German Cancer Aid. The authors would like to acknowledge the American Association for Cancer Research and its financial and material support in the development of the AACR Project GENIE registry, as well as members of the consortium for their commitment to data sharing. Interpretations are the responsibility of study authors.

## Author contributions

J.F.C. supervised this work. C.J.B., A.K.S. and J.F.C. wrote the manuscript with support from all authors. C.J.B., K.M.S., A.S.W., M.Y.K. and C.H. prepared and processed the analyses. S.M.C., J.J.P. and D.A.S. provided samples and clinical data. C.J.B., A.K.S., K.M.S. J.A.D., J.J.P., D.A.S. and J.F.C. performed data analysis and interpretation.

## Competing interests

J.A.D. is a co-founder of Caribou Biosciences, Editas Medicine, Intellia Therapeutics, Scribe Therapeutics, and Mammoth Biosciences. J.A.D. is a scientific advisory board member of Caribou Biosciences, Intellia Therapeutics, eFFECTOR Thera-peutics, Scribe Therapeutics, Synthego, Metagenomi, Mammoth Biosciences, and Inari. J.A.D. is a Director at Johnson & Johnson and has sponsored research projects by Pfizer, Roche Biopharma, and Biogen. J.F.C. is a co-founder of Telo Therapeutics, Inc. and has ownership interests. The other authors declare no competing interests.
