## [Peer Review File · Nature Communications]

REVIEWER COMMENTS

Reviewer #1 (Remarks to the Author): Expert in glioblastoma genomics

The authors studied functional consequences of rare duplication-insertion in the promoter of TERT in depth. They identified 21 cases of various types of cancer with TERT promoter duplications after exhaustive database search, including 2 GBM cases of their own. Among the 20 possible insertion sites, they found that only the duplication-insertion that creates de novo ETS 200 in addition to the native ETS 200 site increased promoter activity. Knocking down of GABPA resulted in abrogation of upregulated transactivation, indicating that GABP may play a functional role in the activation of TERT duplication. They further showed that TERT duplication was one of the earliest and clonal changes by demonstrating the presence of TERT duplication in all portions of the GBM they examined compared with other changes such as EGFR amplification.

This is a very exhaustive and elaborate study to analyze the function of the rare TERT promoter duplications. TERT duplication have previously been reported in two cases and shown to have increased promoter activity elsewhere. The in-depth functional analysis in relation to GABP is a good extension of the authors' previous study. As TERT promoter mutation is by far the most common somatic mutational events in gliomas and included as a part of the diagnostic criteria in the latest WHO Classification, the result is a significant and important addition to the existing knowledge. The experiments are all sound and appear to be credible.

The limitation of the study is that the frequency of TERT duplication is extremely low, well below 1% of the conventional TERT hotspot mutations (Suppl Fig 1A). Although the duplication should be kept in mind when screening TERT alterations in tumors, particularly in gliomas, the impact on clinical diagnosis may be limited.

I have only a few minor comments:

Fig. 1F is very intriguing but slightly difficult to comprehend at first. If the inserted sequence is shown in bold and its original sequence underlined, it would be easier to understand. A wild type uninserted sequence may also be shown alongside the figure. In the legend for Fig. 1F, -94_-73 = -95_-74, -96_-75, -97_-76, -98_-77 may be added to indicate that those sequences are identical.

It would be useful if they provide the number of GBM searched in the database together with the number of TERT promoter hotspot mutations and duplications. This would provide an idea about the frequency of TERT alterations in GBM.

Reviewer #2 (Remarks to the Author): Expert in glioblastoma genomics and computational genomics

The authors identify a duplication in wild type TERT promoter sequence that, like a well-known clonal oncogenic single point mutation, appears to cause upregulation of TERT expression. The authors go on to show that the duplication only has this effect when it is introduced in phase to enable a novel binding site motif and likely recruitment of a specific TF (the same mechanism for the aforementioned oncogenic mutation). The authors show that this is a clonal event in patient samples.

I found this to be an enjoyable and interesting publication with clear clinical impact: TERTp mutation status is used in clinical practice but, as yet, duplications events are not investigated. The duplication may be rare, and the full mechanism needs to be fully validated but this work provides information that could be immediately relevant to clinical practice and will likely be the impetus for researchers working on TERTp based therapeutic strategies to include these additional genomic events in their experiments.

I only have some minor comments with regards corrections

1. A figure would help the reader to digest the info on the last 9 lines on page 3 (heterotetramer/heterodimer explanation). In act 'Figure 6' is never cited in the manuscript but the top part of it could be exactly what is needed here as Figure 1
2. "Table 1 includes 21 duplications we identified that are contained within the core region of the TERTp, including six samples from four patients in our cohort"
Three of these are from the same patient i.e. two from same primary and one from the recurrence. This just means it was clonal, as shown later on, so this is misleading with regards true population prevalence
3. SuppFig2 – what is the meaning of black, blue and red bars in B and C
4. "Next, we investigated whether GABP can bind to the fragment of TERTp with the 22bp insertion in an electromobility shift assay. Our results show GABP binds to sequences with duplicated native ETS sites (Figure 5, Supplementary Fig 3)." Figure 5?! Surely this should be Figure 3. Supp F3 contains the wild

type EMSA images but these need to also be in Figure 5(3?) so we can see how the insertion alters shifts binding compared to the wild type

5. Pyclone should be PyClone

6. "TERT mRNA positive cells were observed in a second glioblastoma case with a TERTp duplication (Supplementary Figure 6A,B)." Should be Supp Fig 7

7. Figure 4 (E-G) needs to be in bold so readers can see that what follows is the description of those panels

8. Figure 4 "C# indicates the FFPE tissue block and T# indicates the punch". This key is not accurate, and 'N' is not included. I assume N is the single normal adjacent punch. One normal for comparison is not ideal (could more be added? Might not be possible) and there is no comment on why the GABP subunits seem to have far reduced expression in the tumour than the normal....

9. Reiterated from point 1: Figure 6 does not appear to be referenced

Reviewer #3 (Remarks to the Author): Expert in transcription factors, transcriptional regulation, and functional genomics

Overall, this is a rigorous study describing a mechanism that adds significantly to our understanding of non-coding genetic drivers of human cancer. I have reviewed all of the evidence in this study and find it to be robust and deserving of publication. While the frequency of these duplications is quite low, the evidence is presented is quite compelling that these duplications are an alternative route to achieve elevated TERT expression in cancer, which complements the prior work on de novo ETS point mutations. I believe the evidence provided is suitable for publication.

We sincerely thank the three reviewers for their strongly positive comments, corrections, and excellent recommendations. We were fortunately able to obtain 3 additional punches of adjacent normal brain and 4 additional punches of the tumor (Reviewer 2, #8). We isolated RNA and performed RT-PCR on all samples again to address the paucity (one punch) of adjacent normal brain. We also bolster our discussion of the clinical impact of this study with new text and six additional citations of the cancer types where *TERT* promoter mutation status has clinical significance (Reviewer 1). We feel these changes have further improved the manuscript. Please see our detailed responses below.

Reviewer #1 (Remarks to the Author): Expert in glioblastoma genomics

The authors studied functional consequences of rare duplication-insertion in the promoter of *TERT* in depth. They identified 21 cases of various types of cancer with *TERT* promoter duplications after exhaustive database search, including 2 GBM cases of their own. Among the 20 possible insertion sites, they found that only the duplication-insertion that creates de novo ETS 200 in addition to the native ETS 200 site increased promoter activity. Knocking down of GABPA resulted in abrogation of upregulated transactivation, indicating that GABP may play a functional role in the activation of *TERT* duplication. They further showed that *TERT* duplication was one of the earliest and clonal changes by demonstrating the presence of *TERT* duplication in all portions of the GBM they examined compared with other changes such as EGFR amplification.

This is a very exhaustive and elaborate study to analyze the function of the rare *TERT* promoter duplications. *TERT* duplication have previously been reported in two cases and shown to have increased promoter activity elsewhere. The in-depth functional analysis in relation to GABP is a good extension of the authors' previous study. As *TERT* promoter mutation is by far the most common somatic mutational events in gliomas and included as a part of the diagnostic criteria in the latest WHO Classification, the result is a significant and important addition to the existing knowledge. The experiments are all sound and appear to be credible.

The limitation of the study is that the frequency of *TERT* duplication is extremely low, well below 1% of the conventional *TERT* hotspot mutations (Suppl Fig 1A). Although the duplication should be kept in mind when screening *TERT* alterations in tumors, particularly in gliomas, the impact on clinical diagnosis may be limited.

Author response: We are thankful for the many positive comments, particularly that our study is an important addition to the existing knowledge, the experiments are all sound and appear to be credible. We have the following thoughts regarding the statement that due to the low frequency of the duplication, the impact on clinical diagnosis may be limited. The frequency of the *TERT* duplication seems to be low based on our search in the UCSF500 and GENIE data sets. However, at this time an accurate frequency estimate is confounded by poor and variable (among algorithms) detection of insertions or deletions of this size and by the extremely GC rich sequence of the *TERT* promoter. On the other hand, we think it is very important for clinicians treating a variety of cancer patients to be aware of the driver role of the *TERT* duplication for tumor classification and treatment planning. Many of the computational pipelines used for clinical sequencing data will require adjustment specifically in relation to the *TERT* promoter.

Furthermore, detection of the duplication and knowing its driver role is not only relevant in diffuse gliomas, where diagnosis of a histologically low-grade tumor to a high-grade tumor (e.g. a primary GBM) is often followed by more aggressive treatment (PMID: 34148105), there are many more examples in other tumor types where *TERT* promoter mutations can change grading or diagnosis. These examples include *TERT* promoter status in meningiomas (PMID: 34185076), ovarian clear cell carcinoma (PMID: 24338723), small cell carcinoma of the bladder (PMID: 25042800), high-grade adenocarcinoma of the prostate (PMID: 31393284), conjunctival melanoma (PMID: 25159205) and thyroid carcinoma (PMID: 24476079). Our data shows that *TERT* duplications occur in seven cancer types thus far, suggesting their detection will have an impact in many fields of oncology. Also, our key finding that the insertion site and length of the duplication are critical to attract GABP, mimicking the hotspot mutations, further supports a critical role for GABP in *TERT* promoter activation and tumor cell immortality.

REVIEWER: Fig. 1F is very intriguing but slightly difficult to comprehend at first. If the inserted sequence is shown in bold and its original sequence underlined, it would be easier to understand. A wild type uninserted sequence may also be shown alongside the figure. In the legend for Fig. 1F, -94_-73 = -95_-74, -96_-75, -97_-76, -98_-77 may be added to indicate that those sequences are identical.

Author response: As suggested, we bolded the inserted sequence and underlined the original sequence. We added a wild type uninserted sequence at the top. In the Figure 1F legend, we added -94_-73 = -95_-74, -96_-75, -97_-76, -98_-77 to the list of promoter sequences that are identical.

REVIEWER: It would be useful if they provide the number of GBM searched in the database together with the number of *TERT* promoter hotspot mutations and duplications. This would provide an idea about the frequency of *TERT* alterations in GBM.

Author response: We followed the reviewer's suggestion and tallied the number of *TERT*_p hotspot mutations vs duplications in the two databases (UCSF500 and GENIE) we used. The results are included in the manuscript as Supplementary Table 1. However, especially for the GENIE dataset we think the results might not be representative of the population because *TERT*_p alterations, including hotspot mutations, were only detected in one third of their glioblastomas. In several independent studies of GBM, the frequency of *TERT* promoter mutation is over 80%.

Reviewer #2 (Remarks to the Author): Expert in glioblastoma genomics and computational genomics

The authors identify a duplication in wild type *TERT* promoter sequence that, like a well-known clonal oncogenic single point mutation, appears to cause upregulation of *TERT* expression. The authors go on to show that the duplication only has this effect when it is introduced in phase to enable a novel binding site motif and likely recruitment of a specific TF (the same mechanism for the aforementioned oncogenic mutation). The authors show that this is a clonal event in

patient samples.

I found this to be an enjoyable and interesting publication with clear clinical impact: TERTp mutation status is used in clinical practice but, as yet, duplications events are not investigated. The duplication may be rare, and the full mechanism needs to be fully validated but this work provides information that could be immediately relevant to clinical practice and will likely be the impetus for researchers working on TERTp based therapeutic strategies to include these additional genomic events in their experiments.

Author response: We thank the reviewer for calling out so many positive aspects of our study, including the clear clinical impact.

REVIEWER: I only have some minor comments with regards corrections

1. A figure would help the reader to digest the info on the last 9 lines on page 3 (heterotetramer/heterodimer explanation). In fact 'Figure 6' is never cited in the manuscript but the top part of it could be exactly what is needed here as Figure 1

Author response: We followed the suggestion and include a GABP tetramer figure in Figure 1B. Also, we have now cited Figure 6 on page 22 (see point 9 below).

2. "Table 1 includes 21 duplications we identified that are contained within the core region of the TERTp, including six samples from four patients in our cohort"
Three of these are from the same patient i.e. two from same primary and one from the recurrence. This just means it was clonal, as shown later on, so this is misleading with regards true population prevalence

Author response: We have adjusted the text accordingly. It now reads "Table 1 includes 21 samples from 18 cases we identified that are contained within the core region of the TERTp, including six samples from four patients in our cohort."

3. SuppFig2 – what is the meaning of black, blue and red bars in B and C

Author response: We now indicate in the legend for SuppFig2 that the black bars represent the constructs that are positive and negative controls along with the duplication identified in several tumor samples. The other bars indicate constructs for which the ETS sites are moved out of phase and closer together (blue bars) or further apart (red bars) relative to the ETS sites in the duplication in the tumor sample.

4. "Next, we investigated whether GABP can bind to the fragment of TERTp with the 22bp insertion in an electromobility shift assay. Our results show GABP binds to sequences with duplicated native ETS sites (Figure 5, Supplementary Fig 3)." Figure 5?! Surely this should be Figure 3. Supp F3 contains the wild type EMSA images but these need to also be in Figure 5(3?) so we can see how the insertion alters shifts binding compared to the wild type

Author response: Thank you. We fixed the figure numbering and moved the EMSA images to Figure 3.

5. Pyclone should be PyClone

Author response: We changed Pyclone to PyClone throughout the manuscript.

6. “TERT mRNA positive cells were observed in a second glioblastoma case with a TERTp duplication (Supplementary Figure 6A,B).” Should be Supp Fig 7

Author response: We fixed the figure number, thank you for catching our error.

7. Figure 4 (E-G) needs to be in bold so readers can see that what follows is the description of those panels

Author response: We bolded Figure 4 (E-G) in the Figure legend.

8. Figure 4 “C# indicates the FFPE tissue block and T# indicates the punch”. This key is not accurate, and ‘N’ is not included. I assume N is the single normal adjacent punch. One normal for comparison is not ideal (could more be added? Might not be possible) and there is no comment on why the GABP subunits seem to have far reduced expression in the tumour than the normal....

Author response: We believe the one normal sample in question may be an outlier and certainly had much greater variability in the measurement. Fortunately, we were able to acquire three additional punches from adjacent normal brain tissue (four normal adjacent controls in total) and four additional punches from tumor tissue of the same patient, extracted RNA and repeated the quantitative PCR on all samples. We did detect a range of GABPA and GABPB1 expression in the adjacent normal that resembles the expression variation found in tumor samples, though for each gene tested, adjacent normal brain had lower expression and more outliers. *TERT* expression was not detected in any of the adjacent normal brain tissues as expected. We also edited the Figure legend for more clarity.

9. Reiterated from point 1: Figure 6 does not appear to be referenced

Author response: Thank you for catching this error, we have now cited Figure 6 on page 22 “The length of the *TERT*p duplication puts the two ETS motifs in phase and at a distance that is most ideal for tetramer binding (Figure 6).”

Reviewer #3 (Remarks to the Author): Expert in transcription factors, transcriptional regulation, and functional genomics

Overall, this is a rigorous study describing a mechanism that adds significantly to our understanding of non-coding genetic drivers of human cancer. I have reviewed all of the evidence in this study and find it to be robust and deserving of publication. While the frequency of these duplications is quite low, the evidence is presented is quite compelling that these

duplications are an alternative route to achieve elevated TERT expression in cancer, which complements the prior work on de novo ETS point mutations. I believe the evidence provided is suitable for publication.

Author response: Thank you!

REVIEWERS' COMMENTS

Reviewer #1 (Remarks to the Author):

The authors have satisfactorily responded to my comments. I have no more comments to make.

Reviewer #2 (Remarks to the Author):

I am happy that the authors have made all of the minor amends and was pleased to see they were able to acquire more tissue punches to further bolster their report.

REVIEWERS' COMMENTS

Reviewer #1 (Remarks to the Author):

The authors have satisfactorily responded to my comments. I have no more comments to make.

Author response: Thank you!

Reviewer #2 (Remarks to the Author):

I am happy that the authors have made all of the minor amends and was pleased to see they were able to acquire more tissue punches to further bolster their report.

Author response: Thank you!